# Noncanonical Roles of RAD51

**DOI:** 10.3390/cells12081169

**Published:** 2023-04-15

**Authors:** Mélissa Thomas, Caroline Dubacq, Elise Rabut, Bernard S. Lopez, Josée Guirouilh-Barbat

**Affiliations:** 1INSERM U1016, UMR 8104 CNRS, Institut Cochin, Université de Paris Cité, 24 rue du Faubourg St. Jacques, F-75014 Paris, France; 2Institut de Biologie Paris Seine, IBPS, Neuroscience Paris Seine, NPS, INSERM, CNRS, Sorbonne Université, F-75005 Paris, France

**Keywords:** RAD51, homologous recombination, double-strand break repair, replication stress, RNA:DNA hybrids, post-replication repair, genome instability, cancer predisposition, Fanconi anemia, congenital mirror syndrome

## Abstract

Homologous recombination (HR), an evolutionary conserved pathway, plays a paramount role(s) in genome plasticity. The pivotal HR step is the strand invasion/exchange of double-stranded DNA by a homologous single-stranded DNA (ssDNA) covered by RAD51. Thus, RAD51 plays a prime role in HR through this canonical catalytic strand invasion/exchange activity. The mutations in many HR genes cause oncogenesis. Surprisingly, despite its central role in HR, the invalidation of RAD51 is not classified as being cancer prone, constituting the “RAD51 paradox”. This suggests that RAD51 exercises other noncanonical roles that are independent of its catalytic strand invasion/exchange function. For example, the binding of RAD51 on ssDNA prevents nonconservative mutagenic DNA repair, which is independent of its strand exchange activity but relies on its ssDNA occupancy. At the arrested replication forks, RAD51 plays several noncanonical roles in the formation, protection, and management of fork reversal, allowing for the resumption of replication. RAD51 also exhibits noncanonical roles in RNA-mediated processes. Finally, RAD51 pathogenic variants have been described in the congenital mirror movement syndrome, revealing an unexpected role in brain development. In this review, we present and discuss the different noncanonical roles of RAD51, whose presence does not automatically result in an HR event, revealing the multiple faces of this prominent actor in genomic plasticity.

## 1. Introduction

Genome stability maintenance is essential for the faithful transmission and functioning of genetic material. Indeed, the genome is routinely assaulted by exogenous as well as endogenous stresses, causing genetic instability that can fuel carcinogenesis and/or ageing [1,2,3]. To cope with genotoxic stresses, cells have developed the DNA damage response (DDR), which coordinates the cell cycle progression with DNA repair. Homologous recombination (HR) is one of the most evolutionarily conserved processes governing genome maintenance and plasticity. The ability to exchange homologous sequences directly derives from the DNA structure of two complementary strands, as proposed by Watson and Crick in the early days of these discoveries [4]. HR plays paramount roles in the equilibrium between the genetic stability and variability. It is essential to maintain genomic stability through its DNA repair functions, but it can also promote genomic variability [5,6,7,8]. Different types of lesions can be repaired using HR, mainly DNA double-strand breaks (DSBs) and interstrand crosslinks (ICLs) [9,10]. The protection and resumption of the arrested replication forks is also an essential role of HR in genomic stability preservation. Of note, the prolonged arrest of the replication fork evolves into the formation of DSBs, which can then be processed by HR (or non-homologous end-joining) [11,12].

The defects in HR lead to genetic instability, and many of the genes controlling HR are mutated in cancer [13,14]. This is particularly true for familial breast and ovarian cancer, in which the most frequent germline pathogenic variants affect the HR genes *BRCA1* and *BRCA2* (and to a lesser extent *PALB2*, *RAD51C,* and *RAD51D*). In addition, mutations in the HR genes are also observed in sporadic cases. Surprisingly, despite its central role(s) in HR (see below), very few mutations of *RAD51* have been described in familial or sporadic cancer cases, but their causality has not been demonstrated [13]. In contrast, the overexpression of RAD51 is frequently found in tumors [13,15,16,17,18,19,20]. Moreover, germline pathogenic variants of several HR genes have been described in subgroups of Fanconi anaemia (FA), a rare autosomal recessive syndrome associated with bone marrow failure, developmental malformations, and predispositions to acute myeloid leukaemia and cancers. The FA cells exhibit spontaneous chromosomal instability, which is exacerbated by an exposure to the agents generating the DNA interstrand crosslinks [21]. The HR gene variants found in FA include *BRCA1* (FA-S), *BRCA2* (FA-D1), *PALB2* (FA-N), *RAD51C* (FA-O), and *XRCC2* (FA-U). Whereas three pathogenic variants in *RAD51* have been reported in four cases of an atypical form of FA (FA-R) (Table 1), no cancer predisposition has been associated with these *RAD51* mutations to date (cases reported at <4 months and 13, 23, and 9 years of age). This contrasts with the other HR genes mutated in FA [22,23,24,25]. Collectively, because RAD51 behaves differently than its mediator/accessory proteins for cancerous predisposition, despite its central role in HR, these data reveal the “RAD51 paradox” [13]. Several hypotheses could account for this paradox. One hypothesis proposes that RAD51 should have essential roles, independent of its mediator/accessory proteins, and whose ablation would be too detrimental for the cells’ survival, impairing their development, and thus, tumor development. These activities should be independent of the RAD51 mediator/accessory proteins and different from the RAD51 canonical homology search/strand exchange activity (which involves the RAD51 mediators and accessory proteins). These additional functions are “noncanonical” but should be essential function(s). Indeed, several publications have revealed that RAD51-binding to DNA has consequences for the protection of the arrested replication forks or the protection against alternative and mutagenic DSB repair processes, such as single-strand annealing (SSA) or alternative end-joining (A-EJ). Moreover, a noncanonical role of RAD51 has been described in the processing of RNA:DNA hybrids (R-loops). Finally, the pathogenic variants of *RAD51* in the congenital mirror movement syndrome (Table 1) revealed an unexpected role of RAD51 in brain development [26,27,28,29].

In this review, we discuss these noncanonical roles of RAD51. First, we will summarize the canonical catalytic function of RAD51. Second, we will discuss how RAD51 protects against highly mutagenic SSA and A-EJ, through DNA occupancy, i.e., independently of its canonical catalytic activity (homology search and strand exchange with a homologous duplex partner). Third, we will discuss the noncanonical roles of RAD51 connected to replication: (i) the reversion and (ii) the protection of the arrested replication forks; (iii) the role in post-replication repair and DNA translesion synthesis through the mono-ubiquitination of PCNA. Then we will present the noncanonical roles of RAD51 in the processing of R-loops (RNA:DNA hybrids), both in their formation and in their removal. Lastly, we will address the unexpected role of RAD51 in the development of the nervous system and the consequences of RAD51 mutations in the congenital mirror movement syndrome.

## 2. Catalytic Role of RAD51 in HR

HR is based on the use of sequence homologies between two interacting DNA molecules. Therefore, the search for sequences homology, and the exchange of the homologous sequences represent the pivotal steps of HR, giving the process its name. DSB repair mediated by HR exemplifies the main steps of HR and is summarized in Figure 1. Note that the central steps, i.e., the homology search and strand exchange, are promoted by the filament formed by RAD51 loaded onto the resected ssDNA, which represents the actual “active species” of HR. RAD51 is conserved from prokaryotes (*RecA*) to eukaryotes (*RAD51*), in contrast to many of its partners that act as RAD51 mediators in HR, such as BRCA1, BRCA2, or PALB2, whose phylogenetic conservation is more limited. The role of the RAD51/ssDNA filament in the homology search and strand exchange is also conserved during evolution and can, therefore, be considered the “canonical” role of RAD51.

## 3. Noncatalytic Roles of RAD51 in DSB Repair

DSBs are generally considered the most toxic DNA lesions and can generate genomic instability. DSBs can originate from exogenous (ionizing radiation) as well as endogenous stresses (reactive oxygen species, ROS; replication stress). In addition, the induction of cell-controlled DSBs is produced during physiological processes aimed at generating genetic variability, such as meiosis or the establishment of the immune repertoire. Cell-controlled DSBs in the transcription promoters have also been shown to control the expression of the genes in neurons [37].

Cells employ two main strategies for repairing DSBs (Figure 2). The first strategy is based on the use of an intact homologous sequence and is referred to as HR (Figure 1). The second strategy joins the DNA ends without requiring homologous sequences to repair DSBs and is referred to as nonhomologous end-joining (NHEJ). Contrary to what is frequently written, KU/ligase 4-dependent canonical NHEJ (C-NHEJ) is not error prone. Instead, it is very efficient in joining cohesive or blunt DNA ends without modifying them. However, C-NHEJ is also adaptable and can ligate short noncomplementary ssDNA ends [38,39]. In fact, processing factors (nucleases, non-template polymerases) are responsible for mutagenic outcomes at the repair junctions. Moreover, other pathways, such as alternative end-joining (A-EJ, also called alt-NHEJ, MMEJ, or B-NHEJ) or single-strand annealing (SSA), which are exclusively nonconservative, can repair the DSBs, altering genome stability maintenance (Figure 2).

The choice of the appropriate DSB repair pathway is essential for the balance between the genomic stability and variability. The selection of the DSB repair pathway operates in two steps (Figure 2), as we have proposed [38,40]: First, a choice between C-NHEJ and resection of the DNA ends; second, a choice between HR and the mutagenic repair processes SSA or A-EJ.

**Figure 2 cells-12-01169-f002:**
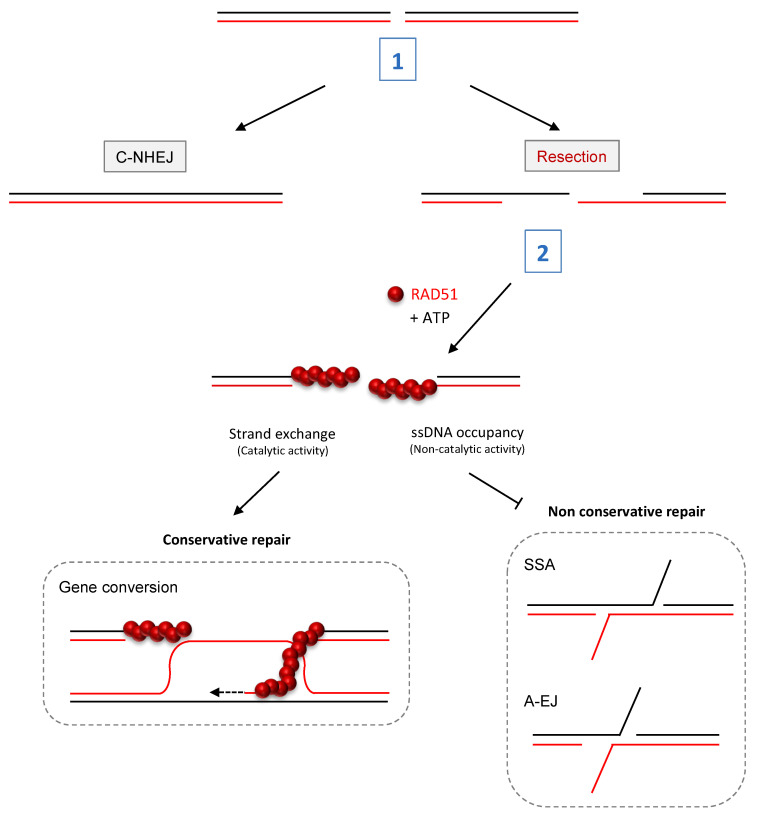
The selection of the DSB repair pathway in two steps. **1.** Competition between end-joining of the DSB through the KU70/KU80/ligase 4-dependent NHEJ pathway, called canonical NHEJ (C-NHEJ), *versus* the resection of the DNA ends generating 3’-ssDNA tails. **2.** On resected ssDNA, competition between conservative HR *versus* alternative nonconservative SSA or A-EJ. SSA and A-EJ occur upon the annealing of the two complementary ssDNA generated by the resection. SSA acts on long homologous sequences, whereas A-EJ acts on microhomologies (few bp). Since both processes anneal complementary sequences that can be distant from the DSB, they both ineluctably lead to the loss of the intervening sequence and are, thus, nonconservative. RAD51 (red dots) not only favors the gene conversion through its catalytic activity but also prevents SSA and A-EJ by occupying the ssDNA and impairing the annealing of the complementary strands [41].

Therefore, at the post-resection step, the maintenance of the genomic stability requires the fostering of HR and the protection against the nonconservative mechanisms SSA and A-EJ. At this second step, the loading of RAD51 on the ssDNA favors conservative HR. Consistently, the suppression of RAD51 or BRCA2 (which loads RAD51 on ssDNA) leads to the stimulation of SSA and A-EJ [32,41,42,43]. In agreement with this concept, the inactivation of HR in breast tumors leads to the accumulation of deletions (with an increased use of microhomologies), which are considered hallmarks of the increased activity of A-EJ [44]. Moreover, the repression of the strand exchange activity of RAD51 results in increased mutagenesis upon UV exposure, likely through the channeling toward the DNA Traslension synthesis (TLS, see below) for the bypass the UV damages [45]. Base substitutions are also characteristic of BRCA-mutated tumors, potentially as a consequence of the increased TLS when replication stalling is not manageable by HR [44,46].

However, BRCA2-deficient cells can promote homology-directed DSB repair (HDR) in a RAD52-dependent manner, and synthetic lethality between BRCA2 and RAD52 has been reported [34,35]. Moreover, the inhibition of HR is not automatically balanced by the increase in the SSA and A-EJ. Indeed, the expression of a dominant negative form of RAD51 (SMRAD51) that compromises HR but is able to bind to ssDNA does not permit the stimulation of A-EJ and SSA [41]. This reveals a separation of function for RAD51 between HR catalysis and prevention of the nonconservative A-EJ and SSA. RAD51 might act at two steps for the A-EJ and SSA repression, namely at the resection step, as shown in the protection of the arrested replication forks (see below), and/or at the annealing of complementary strands, which is required for both A-EJ and SSA (Figure 2). In contrast to arrested replication forks, RAD51 does not protect against extended resection on DSBs generated by endonucleases, suggesting that the two processes are differently regulated [41]. In an in vitro assay that monitored the annealing of complementary ssDNA without the possibility of strand exchange, the binding of wild-type RAD51 (HR proficient) as well as of SMRAD51 (HR deficient) prevented the annealing of the complementary ssDNA [41]. This indicates that RAD51 protects against A-EJ and SSA independently of its catalytic (canonical) strand exchange activity, through a non-catalytic occupancy of the ssDNA, preventing the annealing of the complementary strands [41].

HR is restricted to the S and G2 phases of the cell cycle, when the sister chromatid is available to guarantee an error-free repair. Nevertheless, the loading of RAD51 onto DNA in G1 was recently described at centromere breaks. This loading prevents centromeric instability and chromosomal rearrangements mediated by SSA and POL theta-mediated A-EJ [47]. In the same manner, RAD51 was shown to be recruited at centromeres in quiescent RPE-hTERT cells and to protect centromeres from breakage in a way that requires the strand exchange activity [48].

Therefore, RAD51 can protect genome integrity according to two processes: (i) for the selection of the DSB repair process at the second step, RAD51 channels the DSB repair toward HR, which is conservative, through its canonical strand exchange activity; (ii) additionally, RAD51 prevents nonconservative SSA and A-EJ by impairing the annealing of the complementary ssDNA through the DNA occupancy, independently of its strand exchange activity, i.e., in a noncanonical way.

Recently, it has been shown that RAD51 was implicated in the protection of T cells against senescence, promoting long term immunological memory [49]. Indeed, antigen-presenting cells (APC) were found to transfer telomeres sequences to T cells, maintaining the telomere length of the recipient T cells, and thus, protecting them from senescence. This telomere transfer mechanism acts through an intercellular process that requires synapses between the APCs and T cells and the production of extracellular vesicles by the APCs. These vesicles contain telomeres fragments and the HR proteins BRCA1, BRCA2, and RAD51. The production of the telomeres-containing vesicles is independent of RAD51 but the content of the telomeric ssDNA in the vesicles, and the lengthening of the telomeres in the recipient T cells require RAD51 [49]. Deciphering the respective parts of the catalytic strand invasion activity versus the noncanonical ssDNA protection function by RAD51 in this process remains to be investigated.

## 4. Noncatalytic Roles of RAD51 in DNA Replication

HR proteins are found on normal replicating forks [50]. Stemming from replicative stress, in addition to its functions in HR that help to restart the collapsed replication forks, RAD51 plays additional roles that are independent of its strand exchange activities.

### 4.1. RAD51 Promotes Replication Fork Reversal in a BRCA2-Independent Manner

When the replication forks stall, one possibility is to promote fork reversal, which involves the annealing of the two nascent strands and the reannealing of the two parental strands (Figure 3). RAD51 is required for the reversal of the fork [51], which creates a four-branch structure called the “chicken foot” (Figure 3) that can migrate to elongate the reversed arm [52,53,54].

Translocases (SMARCAL1, ZRANB3, HLTF, and RAD54) and/or helicases (BLM, FBH1, and WRN/FANCM) foster the migration of the fork and the elongation of the reversed arm [55,56]. RAD51 cannot catalyze fork reversal alone in vitro [57], although it is capable of strand exchange, suggesting that the two processes are mechanistically different. 

Replication fork reversal occurs in *BRCA2 KO* cells [58,59,60], showing that BRCA2 is not necessary for the RAD51-mediated fork reversal. Instead, the recruitment of RAD51 on the forks was suggested to be mediated by the direct interaction with DNA polymerase alpha [61], RAD54 [57], or RAD51C [62,63]. Additionally, MCM8 and MCM9, which are implicated in the pre-replication complex (pre-RC) formation, DNA replication elongation and aid in the normal progression of the replication fork, favor the recruitment of BRCA1 and RAD51 when the fork stalling occurs, to protect them from excessive degradation [64]. One hypothesis is that the ssDNA tracts are relatively short, thus no mediator (such as BRCA2) is required to remove RPA from them. Consistent with that observation, fork reversal occurs in cells expressing a mutant form of RAD51 that was described in a Fanconi patient, RAD51-T131P [60]. This mutant does not form RAD51 foci when DNA damage occurs, which seems to confirm that long and stable filaments of RAD51 are not required for replication fork reversal [25,41]. Consistent with a dissociation of function between strand exchange (canonical role) and fork reversal activities, a mutant form of RAD51 that is unable to catalyze strand exchange, RAD51-II3A, is capable of promoting fork reversal [31].

### 4.2. RAD51 Protects against Extensive Resection of Blocked Replication Forks

A reversed fork can be the target of extensive degradation initiated by MRE11 and EXO1/DNA2. BRCA2 loads RAD51 on the reversed arm to protect the fork against such degradation [63,65]. BRCA2 contains two types of RAD51-binding domains (several BRC repeats and a C-terminal site) and a DNA-binding domain (DBD). The BRC repeats are required for HR, whereas the C-terminal domain of BRCA2 is not. In contrast, the interaction of RAD51 and the C-terminal domain of BRCA2 is absolutely required for the fork protection, and the BRC repeats are not sufficient [65]. In addition, the protection of the reversed forks is independent of the BRCA2 DBD [63,65]. In accordance with the different mechanisms of the BRCA2–RAD51 interplay in HR and the fork protection, the depletion of RADX, which is a negative regulator of RAD51 through its binding onto ssDNA, restores the fork protection but does not restore HR in BRCA2 KO cells [66]. This supports a separation of function between HR and the fork protection.

In addition to BRCA2, BRCA1 and PALB2 are also required to promote RAD51′s function in HR. BRCA1 promotes HR by fostering DNA end resection. Then, BRCA1 directly interacts with PALB2 and recruits BRCA2/RAD51 to the DSB sites. BRCA1, dimerizing with BARD1, enhances the recombinase activity of RAD51 [67]. While the BRCA1/BARD1 interaction is still required to foster the recruitment of RAD51 onto blocked forks, the interaction between BRCA1 and PALB2 is dispensable. Some *BRCA1* mutations and mutations at the interface between BARD1 and RAD51 decrease the protective effect of RAD51 on the blocked replication forks without altering HR [68], supporting the separation of function.

Reports about the mutant forms of RAD51 also confirmed the separation of function between HR and the fork protection. The Fanconi variant of RAD51, RAD51-T131P, is unable to protect the replication forks but is capable of HR when mixed with the endogenous wild-type allele [25,60]. Reciprocally, RAD51-II3A, which forms filaments but is deficient in strand exchange, protects against resection [31]. Finally, overexpression of the RAD51-K133R mutant protein, defective in HR, makes the forks resistant to degradation in BRCA2-deficient cells [65].

The study of these mutants confirmed that the proficiency in strand exchange and protection of the reversed forks are two separate functions of RAD51.

### 4.3. Nonrecombinogenic Functions of RAD51 in Post-Replicative Repair

In addition to its role in replication fork reversal and protection, RAD51 also plays roles in post-replicative repair. Two distinct post-replicative repair modes exist: an error-prone mode involving translesion synthesis (TLS) and a recombination mode, known as template switching (TS). The switch between TLS and TS is determined by mono- or poly-ubiquitination of PCNA, respectively. In TS, one newly synthesized strand serves as a replication template for the other blocked nascent strand. This indicates a strand exchange catalytic activity of RAD51, but in contrast with DSB-induced HR, TS is not initiated by a 3′-double stranded end, but rather by an ssDNA gap [69] (Figure 4).

RAD51 is also implicated in post-replicative repair by TLS, and this is not related to its strand exchange activity [70]. TLS polymerases are recruited onto damaged DNA following mono-ubiquitylation of PCNA at lysine 164 by RAD18. In yeast, Rad51, Rad52, and Rad57, but not Rad54, facilitate Rad6/Rad18-binding to the chromatin and the subsequent DNA damage-induced PCNA ubiquitylation [71]. In human cells, RAD51 physically interacts with FANCD2 and RAD18 in cells treated with hydroxyurea, an inhibitor of the ribonucleotide reductase that generates the replication stress through the imbalance of the nucleotide pool. This RAD51/FANCD2/RAD18 complex favors PCNA monoubiquitylation and the chromatin recruitment of the TLS polymerase POL theta. This function is conserved in the absence of BRCA2 or in the presence of the RAD51 inhibitor B02, showing that this function is independent of the strand exchange activity of RAD51 [72].

Non-recombinogenic activities of RAD51 in post-replicative repair were also observed in combination with the MCM helicase complex. In yeast, MCM helicase physically interact with the recombination proteins Rad51 and Rad52. These interactions occur in a fraction of the chromatin enriched in replication and repair factors [73,74], outside of replication origins and forks [75]. These interactions are detected in G1 and are lost in S/G2 unless replicative lesions are present. In the case of replicative stress, the MCM/Rad51 complex remains bound to the DNA and promotes ssDNA gap repair and replication fork progression. A mutant of Rad51 impaired in its ability to bind to MCM is partially defective in the ssDNA gap filling and replication fork progression through damaged DNA, but is fully proficient in the recombination, suggesting that this function is independent of the recombinogenic activities of Rad51 [75]. The mechanism by which Rad51/MCM acts in post-replicative repair remains elusive. One possibility is that these interactions facilitate the recruitment of Rad6/Rad18 by Rad51/MCM at the sites of replication stress, PCNA ubiquitylation by Rad6/Rad18, and the recruitment of TLS polymerases to fill in the ssDNA gaps [72]. Although not yet demonstrated, these functions might be conserved in mammals, as the MCM–RAD51 interaction is conserved [76].

Therefore, again RAD51 ensures the backup of the replication with both its canonical (strand exchange) and noncanonical (independent of strand exchange) activities. While the restart of the arrested replication forks can involve the canonical activity of RAD51, it is also implicated in the reversion and protection of the arrested replication forks through its DNA-binding activity, but independently of its strand exchange activity. At the post-replication stage, RAD51 can promote template switching through the strand exchange (canonical activity), but also fosters TLS through the mono-ubiquitination of PCNA in a noncanonical way.

## 5. RAD51 and R-Loops: A Double-Edged Relationship

RecA, the bacterial orthologue of RAD51, has been known to promote RNA:DNA hybridization [77,78,79], and *in vitro*, the R-loop reaction is dependent on the RecA protein for optimum efficiency [80]. R-loops are nucleic acid structures composed of an RNA:DNA hybrid and a displaced ssDNA. Interestingly, the RecA-associated R-loop formation is not dependent on ATP hydrolysis [80], unlike the canonical strand exchange reaction between two DNA molecules.

Despite these early findings in bacteria, the relationship between RAD51 and the other HR proteins and R-loops has only begun to be investigated in eukaryotes. 

### 5.1. RAD51 Promotes R-Loop Formation

R-loops can form either in *cis,* when the nascent RNA transcript anneals to its DNA template strand (co-transcriptional R-loops; Figure 5A), or in *trans,* when the RNA transcript binds to homologous DNA at a distant locus (post-transcriptional R-loops; Figure 5B).

In the yeast *Saccharomyces cerevisiae*, R-loops can naturally and transiently occur in wild-type cells [81,82], and these specific R-loops appear to be RAD51-independent [83], whether in *cis* or in *trans* [84]. However, in mutants defective for RNA biogenesis, the RNA:DNA hybrid formation is dependent on RAD51 and RAD52. These hybrids, which are transcription dependent, lead to chromosome loss and terminal deletions of chromosomes [85]. Additionally, these RAD51-dependent R-loops can form both in *cis* and in *trans*, although the *trans* R-loops do not trigger as much genomic instability as their *cis* counterpart. In *Schizosaccharomyces pombe*, it has been shown that Nrl1, a spliceosome-associated protein, can counteract the formation of R-loops driven by Rad51 and Rad52 [83]. This suggests that the RNA:DNA hybrid-forming ability of RAD51 might be heavily context dependent. 

Srs2, a helicase that can remove Rad51 from ssDNA, can antagonize the hybrid-forming ability of Rad51. More specifically, it appears that Srs2 is particularly important for limiting the Rad51-dependent R-loop formation at the highly transcribed rDNA locus [85].

Exposure of human cells to B02, a small molecule that inhibits the strand exchange activity of RAD51 [86], led to an increase in RAD51-dependent R-loop formation, supporting the idea that the strand exchange activity of RAD51 is not necessary for its RNA:DNA hybrid-forming ability [87], thus pointing towards a noncanonical role of RAD51 in R-loop formation.

TERRA stands for telomeric repeat-containing RNA. It is a long non-coding RNA transcribed from chromosome ends that forms R-loops at telomeres [88], preferentially short ones. Its role is to regulate telomere structure and maintenance [89,90]. In human cells, the recruitment of TERRA onto telomeres is promoted by RAD51, while BRCA2 does not appear to play a key role in this process [91] (Figure 6). This suggests that the role of RAD51 in this mechanism is noncanonical. However, RAD51-II3A, a mutant RAD51 protein with no strand invasion activity, cannot drive the recruitment of TERRA onto telomeres [91], calling this hypothesis into question. It is also possible that the mutated site in RAD51-II3A, which is involved in the recognition and pairing of the homologous region [92,93], is necessary for RNA binding. Additionally, while RAD51 does not bind U1 small nuclear RNA *in vivo*, it binds TERRA with a 3-fold higher affinity than homologous telomeric DNA *in vitro* [91]. This suggests that RAD51 might not only bind to a specific subset of RNA but would also do so with a higher affinity than DNA.

Studies in yeast and human cells showed that RAD51 drives the formation of R-loops in specific contexts. However, the relationship between RAD51 and R-loops is twofold. Indeed, RAD51 is also involved in the resolution of R-loops.

### 5.2. RAD51 Promotes the Removal of R-Loops and Counteracts Their Deleterious Effects

Reactive oxygen species (ROS) generate replication stress and induce HR [94]. ROS also strongly induce the R-loop formation at the transcription sites. In human cells, these R-loops are detected by a complex composed of RAD51, RAD52, and Cockayne syndrome protein B (CSB) [95]. Although both RAD51 and RAD52 individually show an affinity for ROS-induced R-loops in vivo, CSB drives the recruitment of the complex at the R-loop sites, while RAD51 plays a key role in dissolving the R-loops, independent of BRCA1/2 and the classical HR pathway [95]. However, ROS-induced R-loops at telomeres do not recruit RAD51. Instead, they favor a CSB/RAD52/POLD3 complex [96]. In wild-type human cells, transcription–replication conflicts (TRCs) trigger the co-transcriptional R-loop formation [97,98]. One of the earliest steps in resolving these TRC-induced R-loops is the recruitment of RAD51 [99]. Additionally, defects in ATR-CHK1 checkpoint signaling also lead to an accumulation of co-transcriptional R-loops [100]. Such defects can be found in Werner syndrome cells, which lack the WRN protein [101]. In the absence of WRN, the WRN-interacting protein WRNIP1 is critical for removing these co-transcriptional R-loops triggered by defects in the ATR-CHK1 signaling. This is achieved by recruiting RAD51 [102].

ROS-induced R-loops and R-loops triggered by transcription–replication conflicts appear to be resolved by RAD51 [95,99,102] independently from the canonical role of RAD51 in repair. Indeed, ROS-induced R-loops are thought to be resolved through a CSB/RAD52/RAD51-mediated pathway but independent of BRCA1/2 [95]. As such, these studies point towards a noncanonical role of RAD51 in removing R-loops, although additional studies are needed to confirm this. TRC-induced R-loops seem to eventually be resolved through replication fork reversal [102] or MiDAS [99], a DNA repair pathway occurring during mitosis.

R-loops have a dual nature. They participate in a number of physiological processes, such as repair and gene expression, but they are also powerful drivers of genomic instability [103,104]. As such, R-loop formation and removal must be tightly regulated, and RAD51, which promotes both their formation and their removal, is emerging as one such important regulator.

Of note, a connection to R-loops has been proposed in several human diseases, including cancer and Fanconi anemia [105]. Since RAD51 can be over expressed in tumors and pathogenic variants of RAD51 have been reported in an atypical form of FA (FA-R), investigating the potential interplay between R-loops, RAD51, and cancer progression or Fanconi anemia should be informative.

RAD51 exercises opposite roles in R-loop management through its noncanonical functions. On the one hand, it favors the R-loop assembly, but on another hand, it fosters the disassembly of R-loops.

## 6. *RAD51* Pathogenic Variants in Congenital Mirror Movement

Congenital mirror movement syndrome (CMM [OMIM #614508]), a rare autosomal dominant neurodevelopmental disease [26,27,28,29], constitutes a pathological situation that might reveal surprising noncanonical roles of RAD51. Indeed, several *RAD51* pathogenic variants have been identified in families with CMM. CMM patients are unable to perform pure unimanual movements and have difficulties performing bimanual asymmetric coordinated movements, such as writing or playing the piano. When intending to perform a voluntary movement with one hand, the other hand performs an involuntary symmetric movement, mirroring the intended movement [106,107,108]. To date, four familial *RAD51* variants have been shown to be involved in CMM (Table 1 and Figure 7): p.R254* (in two families) [28,29], p.P286Tfs*37 [28], p.R250Q [27], and p.T134N [26]. *RAD51* pathogenic variants have an incomplete penetrance and a variable expressivity, which are hallmarks of CMM, regardless of the genetic cause [28,29,109,110,111,112]. The first *RAD51* pathogenic variants associated with CMM were the nonsense p.R254* and frameshift p.P286Tfs*37 variants, both introducing premature termination codons [28]. RAD51-R254* mRNA was prone to nonsense-mediated mRNA decay (NMD), suggesting that *RAD51* haploinsufficiency could explain the pathogenicity of the *RAD51* variants in CMM. However, the more recent discovery of two non-truncating *RAD51* variants (i.e., p.R250Q and p.T134N) [26,27] raised the possibility of alternative pathogenic mechanisms. Molecular pathogenesis may provide insight into the link between the molecular defect and the discrete phenotypes related to human *RAD51* mutations, namely, atypical Fanconi anemia (FA-R) [22,23,24,25], premature ovarian insufficiency [30], and CMM (Table 1 and Figure 7). No cancer predisposition has been reported in the pathologies caused by *RAD51* variants (Table 1).

CMM patients have an altered projection of the corticospinal tract, a major motor tract that sends motor commands from the primary motor cortex area to the contralateral spinal cord, and altered interhemispheric communication, which is necessary for the unilateral activation of the motor cortex during unimanual movement in physiological conditions [113,114]. The bilateral activation of the motor cortices and the higher ratio of uncrossed vs. crossed corticospinal fibers in CMM patients compared to healthy controls may both be involved in the bilateral transmission of the motor command to the spinal cord, accounting for the generation of the mirror movement during the unimanual voluntary movement. Interestingly, *RAD51* is expressed in the corticospinal neurons of neonatal mice during the development of the tract, and the RAD51 protein is detected in the cytoplasm of those neurons and in the soma of the cells when and where their axons cross the midline to form the crossed corticospinal tract [28]. Thus, CMM suggests a new cytoplasmic role for RAD51 in post-mitotic cells during the development of the motor system, possibly in the guidance of the corticospinal axons for crossing the midline. A deeper study is still lacking on the impact of the *RAD51* variants and their expression level to understand the noncanonical role of RAD51 in the development of the corticospinal tract and to unravel the precise molecular pathways that are involved. Whether the cytoplasmic role of RAD51 revealed by CMM depends on the same molecular partners as its other mostly nuclear canonical and noncanonical roles or whether it depends on the still unknown cytoplasmic-specific partners needs further investigation. However, no other HR gene has been implicated in this pathology to date. A functional link between the cyto- or nucleoskeleton and DNA repair proteins, including RAD51, has already been shown in some contexts [115], and a noncanonical role of RAD51 in cytoskeleton dynamics regulation at the axonal growth cone could be a tempting hypothesis. The precise subcellular localization of RAD51 in the corticospinal neurons and the identification of its local partners could open unexplored avenues in the RAD51 roles that are specific to the development of the nervous system.

These data underline a novel, noncanonical role of RAD51 in preventing CMM syndrome. However, this role remains mysterious at the molecular level. Thus, deciphering the molecular role of RAD51 in the development of the nervous system represents an exciting challenge for future prospects.

## 7. Concluding Remarks

*RecA/RAD51* is a highly conserved gene from prokaryotes to eukaryotes, and RecA/RAD51 proteins are well known for their canonical role in the search and exchange of homologous DNA, which is the central step of HR. In recent years, an accumulation of evidence revealed the existence of noncanonical roles of RAD51, which are independent of its strand exchange function between two DNA molecules and/or HR mediators, such as BRCA2. Therefore, the presence/detection of RAD51 in the chromatin should not be automatically interpreted as an HR event.

Remarkably, *RAD51* is marked by a paradoxical absence of cancer predisposition due to altered variants, in contrast to its HR partners (mediators and accessory proteins). One hypothesis that could account for the "RAD51 paradox" suggests that the defect in RAD51 affects not only HR but also its noncanonical roles and that the simultaneous inhibition of all these pathways would be too deleterious for the cell to survive and proliferate. Therefore, the cells bearing one of these *RAD51* mutations would be eliminated, impairing their transformation in tumor cells and *in fine* tumor development.

The DNA-binding capability of RAD51 confers the capacity to protect genome stability, in an independent way of its strand exchange activity. Indeed, RAD51 (i) protects the arrested replication forks against degradation, (ii) prevents excessive breakage at centromeres in G1 or quiescent cells and (iii) prevents nonconservative mutagenic repair (SSA, A-EJ) by inhibiting the annealing of the complementary ssDNA. RAD51 can also favor replication fork reversal and restart and exercise the post-replication functions in noncanonical ways. Notably, the canonical and noncanonical roles of RAD51 are separable, but all of these different roles consistently co-occur for the maintenance of the genomic stability.

The binding of RAD51 to DNA protects against mutagenic repair processes. The ablation of the mediator proteins, such as BRCA2, leads to the inability to load RAD51 onto damaged DNA. Therefore, not only is HR inhibited, but the damaged DNA becomes accessible to alternative mutagenic events. This raises the question of whether the sole alteration of HR is sufficient to promote cancer or whether additional genetic instability generated by the unleashed alternative repair processes is needed. In the latter hypothesis, the alteration of the RAD51 loaders (BRCA2 and PALB2) should favor tumorigenesis, while the RAD51 variants that affect the HR activity but not its binding to DNA should not confer a predisposition to cancer.

In recent years, RNA:DNA hybrids have attracted the attention of the scientific community. RAD51 appears to be involved in both their formation and their resolution, underlying its duality. These RAD51-mediated processes between DNA and RNA can be considered noncanonical roles, as the canonical role of RAD51 involves two DNAs. However, both RNA and DNA are nucleic acids sharing many common structures, and RAD51 might also promote the strand exchange between homologous sequences of RNA and DNA in a canonical-like way. In a mirror fashion, the proteins involved in the RNA metabolism (splicing, transport) are capable of promoting the D-loop between two DNA molecules [116,117,118,119]. However, in several situations, the processing of the RNA:DNA hybrids by RAD51 is separable from its strand exchange activity and is independent of the RAD51 mediators. Since one role of BRCA2 is to remove RPA from ssDNA while RNA is not coated with RPA, the RNA:DNA hybrid formation might be, at least, partially BRCA2-independent. These processes appear to be connected to RNA biogenesis, and many future investigations are needed to establish a clear view of the roles of RAD51 in RNA:DNA hybrid metabolism.

While *RAD51* mutations do not seem to be responsible for oncogenesis, germline pathogenic variants have been described in CMM syndrome (which is not associated with a cancer predisposition), revealing an unexpected, post-mitotic, and probably cytoplasmic role of RAD51 in brain development. Again, future investigations should elucidate this mysterious role of RAD51, which might be independent of its DNA strand exchange activity and/or its classical HR partners. In addition, the post-mitotic role of RAD51 in CMM, in addition to the protection of centromeres recently discovered in quiescent and G1 cells, demonstrates that the roles of RAD51 are no longer restricted to the S/G2 phases of the cell cycle but extend to G1 or even quiescent cells.

RAD51 is a pharmacological target, and active research is performed to develop RAD51 inhibitors [120]. Consequently, considering the impact of these inhibitors on the noncanonical RAD51 functions should be highly valuable. The molecules that can affect all the RAD51 functions, or in contrast, that are able to selectively inhibit one or the other of the different functions of RAD51, should enlarge the panel of therapeutic strategies and to adapt them to different clinical cases.

Together, these data shed light on the multiple faces of a paramount protagonist of genome plasticity that is conserved through the evolution and involved in numerous fundamental biological processes.

## Figures and Tables

**Figure 1 cells-12-01169-f001:**
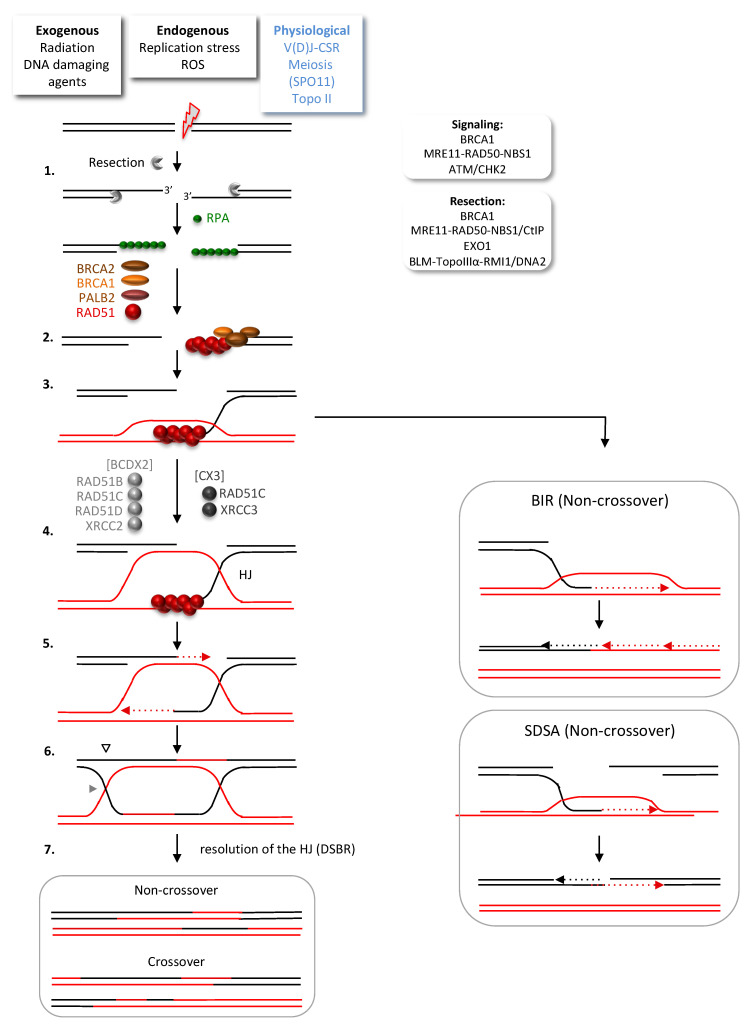
HR-mediated double-strand break repair. Different sources, exogenous, endogenous, or physiological, can generate DSBs, which are signaled by MRE11/RAD50/NBS1 (MRN) and the axis of kinases ATM/CHK2. 1. The resection of the DSB generates 3’-single-stranded DNA (ssDNA), which is coated and protected by the replication protein RPA. The resection is initiated by MRN/CtIP then EXO1 or the association of the helicase BLM (in complex with TopoIIIα and RMI1) with the endonuclease Dna2. 2. BRCA2/PALB2 in mammals (RAD52 in yeast) replace RPA with RAD51, which forms a structured filament. The BRCA2-deficient cells can promote the homology-directed DSB repair in a RAD52-dependent manner [34,35]. 3. The RAD51/ssDNA filament scans the intact duplex DNA, searching for the sequence homology. 4. When sequence homology is found, the RAD51/ssDNA filament promotes strand invasion/exchange, generating a D-loop (displacement loop). 4. The double-strand break repair model (DSBR) [36] implies the capture of the second end. The paralogues of RAD51, which operate in two distinct complexes in mammalian cells, RAD51B-RAD51C-RAD51D-XRCC2 [BCDX2] and RAD51C-XRCC3 [CX3], foster the assembly and stabilization of the ssDNA/RAD51 filament and the HR intermediates. They can also participate in the steps downstream of the homology search. 5. The DNA synthesis primed on the invading 3’-ssDNA copies the complementary sequence (dotted arrows). This generates the HR intermediates bearing the cruciform Holliday junction (HJs). The extension of the DNA synthesis favors the migration of the HJs, thus extending the heteroduplex. 6. The resolution of the HJs completes the repair of the DSB. 7. Depending on the sense of the resolution (triangles), the process will result in a gene conversion (non-reciprocal transfer of the genetic information) associated or not with crossing over (reciprocal exchange of the adjacent sequences). Right panels: in BIR (break-induced replication), the D-loop is stabilized, and DNA synthesis primed by the invading strand can process long distances (even up to the chromosome end). In SDSA (synthesis-dependent strand annealing), the D-loop is not stabilized, and the invading synthetizing strand is rejected and then annealed to the parental DNA strand. These two cases lead to a gene conversion not associated with crossing over.

**Figure 3 cells-12-01169-f003:**
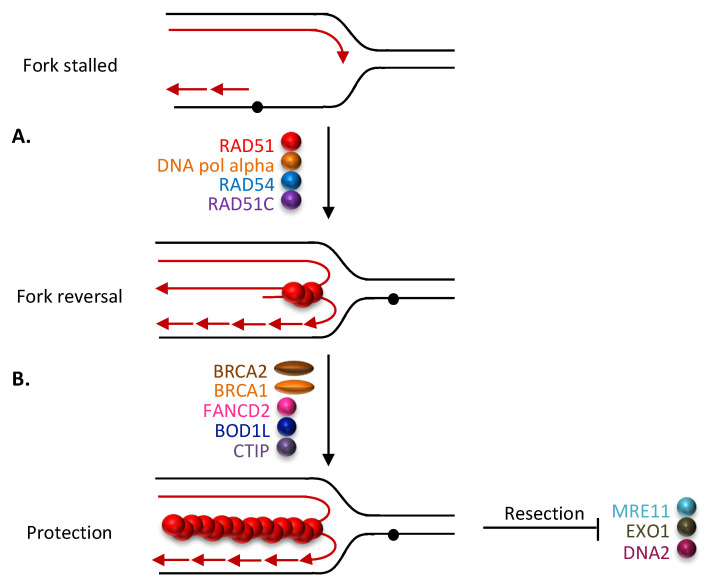
RAD51 and replication forks. (**A**). RAD51 is required for the reversal of the fork. The recruitment, mediated by direct interaction with DNA pol alpha, RAD54 and/or RAD51c creates a four-branch structure called « hicken foot». BRCA2 is not needed at this stage. (**B**). RAD51 protects against the resection initiated by MRE11 and then EXO1/DNA2. Moreover, BRCA2, BRCA1, FANCD1, BOD1 and also CTIP are required for fork protection.

**Figure 4 cells-12-01169-f004:**
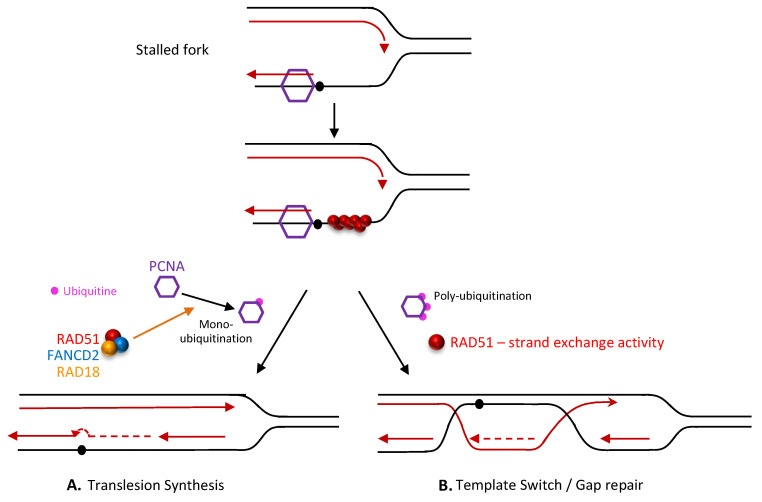
RAD51 and post-replicative repair. The parental strands are shown in black and the nascent strands are shown in red. Damage to the DNA matrix (black dot) can block the progression of the replication (red arrows), generating a single-stranded sequence. RAD51/FANCD2/RAD18 foster the mono-ubiquitination (pink dot) of PCNA (purple hexagon). (**A**). The mono-ubiquitination of PCNA favors DNA synthesis (red dotted line) through the damage using specialized TLS polymerases (TLS: translesion). (**B**). Poly-ubiquitination of PCNA favors the RAD51-mediated strand invasion of the sister chromatid (TS: template switching) allowing to initiate DNA synthesis (dotted line). TS is not initiated by a 3′-double-stranded end but by an ssDNA gap. The RAD51 strand exchange activity is involved in this mechanism and a newly synthesized strand serves as a replication template for the blocked strand.

**Figure 5 cells-12-01169-f005:**
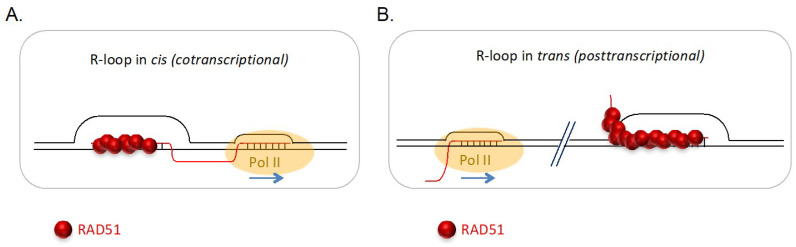
Co-transcriptional and post-transcriptional R-loops form upon hybridization of a nascent RNA transcript (red) with DNA (black). (**A**) Co-transcriptional R-loops form in *cis* when the nascent RNA binds to its DNA template upstream of the migrating transcription bubble. (**B**) Post-transcriptional R-loops form in *trans* when the nascent RNA detaches from its transcription site and then binds to a homologous and distant locus on the DNA strand. Blue arrows: progression of transcription. Red spheres: RAD51 protein.

**Figure 6 cells-12-01169-f006:**
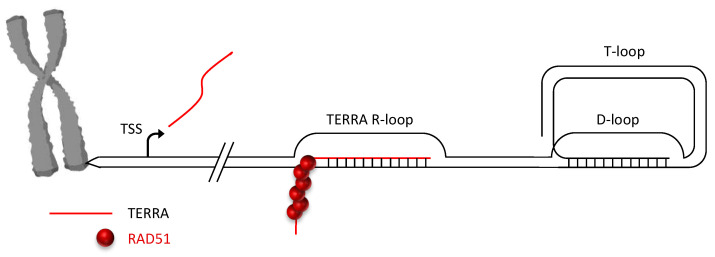
Nucleoprotein structure of the chromosome ends. Telomeres have a 3ʹ overhang, which can invade the dsDNA region, forming a D-loop termed the T-loop. Telomeres are transcribed into TERRA (red line). The TERRA transcription starts in sub-telomeric regions, extending towards the chromosome ends. The TERRA association and R-loop formation at telomeres depend on RAD51, which binds the TERRA and catalyze the TERRA R-loop formation *in vitro*. Red spheres: RAD51 protein.

**Figure 7 cells-12-01169-f007:**
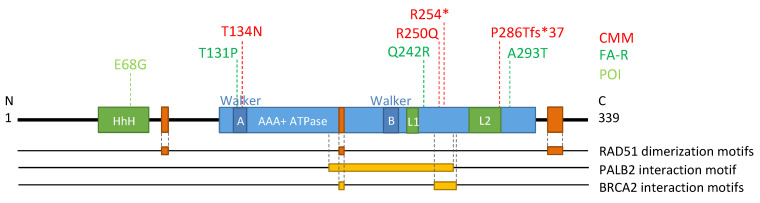
Human RAD51 is a 37 kDa protein (339 amino acids) with an N-terminal domain (1–84) and a C-terminal domain (89–339). The green boxes represent the DNA-binding domains: helix hairpin helix (HhH), L1 loop (L1), and L2 loop (L2). The blue box represents the AAA+ ATPase domain with Walker A and Walker B motifs (dark blue boxes), allowing for the interaction with ATP. The orange boxes represent the RAD51 dimerization motifs: the N-terminal and central motifs of two adjacent protomers interact together; the C-terminal motif (called the “ATP cap”) interacts with the ATP pocket of the adjacent protomer. The yellow boxes represent interaction motifs with the RAD51 partners: the PALB2 interaction motif overlaps with the BRCA2 interaction motifs; the most central BRCA2 interaction motif is identical to the central RAD51 dimerization motif; and the other BRCA2 interaction motif forms a secondary interface. The locations of the pathogenic variants associated with congenital mirror movement (CMM, in red), atypical Fanconi anaemia (FA-R, in dark green), and premature ovarian insufficiency (POI, in light green) are indicated.

**Table 1 cells-12-01169-t001:** *RAD51* pathogenic or experimental variants (restricted to the coding sequence of *RAD51*, excluding the cancer/tumor-associated variants). All the *RAD51* pathogenic variants are heterozygous. Phenotypes: congenital mirror movement (CMM), atypical Fanconi anaemia (FA-R), and premature ovarian insufficiency (POI).

Variants	Cases (inheritance)	Pheno types	Ref.	Domains	Impacts
c.203A>G p.Glu68Gly E68G	Sporadic (unknown)	POI	[30]	HhH domain	Exclusively cytoplasmic; dominant negative
R130A, R303A, K313A “RAD51-II3A”	*Experimental*	/	[31]	DNA binding domain II	HR defective; modest ssDNA binding affinity default; total loss of D-loop activity; defective interaction with the intact dsDNA being scanned
c.391A>C p.Thr131Pro T131P	Sporadic (*de novo*)	FA-R	[25]	Walker A motif in AAA+ ATPase domain	DNA-independent ATPase activity; altered ATP binding and hydrolysis; dominant negative
K133R	*Experimental*	/	[32]	Walker A motif in AAA+ ATPase domain	HR defective; deficient for ATP hydrolysis, reduced DNA binding affinity
c.401C>A p.Thr134Asn T134N	Familial	CMM	[26]	Walker A motif in AAA+ ATPase domain	Not addressed
c.725A>G p.Gln242Arg Q242R	Sporadic (*de novo*)	FA-R	[24]	PALB2 interaction domain	Not addressed (except hypersensitivity to mitomycin C of patient’s cells); suggested to be dominant negative
c.749G>A p.Arg250Gln R250Q	Familial	CMM	[27,33]	BRC interacting motif 2 Nuclear export sequence	Not addressed for HR; suggested loss-of-function in neurons (*in vitro* primary culture)
c.760C>T p.Arg254* R254*	Familial (x2)	CMM	[28,29]	Not applicable (premature termination codon)	Targeted by NMD; haploinsufficiency
c.855dupA p.Pro286Tfs*37 P286Tfs*37	Familial	CMM	[28]	Not applicable (premature termination codon)	Not addressed; probably targeted by NMD; probable haploinsufficiency
c.877G>A p.Ala293Thr A293T	Sporadic (x2) (*de novo*)	FA-R	[22,23]	AAA+ ATPase domain	Impaired DNA-binding, ATPase and strand exchange activities; dominant negative

## Data Availability

Not applicable.

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
