# Peer review of "Noncanonical Roles of RAD51"

_cells, 2023, doi:10.3390/cells12081169_

Round 1

Reviewer 1 Report

The manuscript entitled “Noncanonical Roles of RAD51” by Thomas et al., developed an intriguing and less understood noncanonical roles of RecA/RAD51 in non-conservative mutagenic DNA repair, replication resumption, post-replicative repair, R-loop formation, DNA-RNA hybridization and illustrated about RAD51 pathogenic variants responsible for congenital mirror movement syndrome and fanconi anaemia.

 Specific Concerns: 

1.                    Does RAD54 (RAD54L in human) and other RAD52 epistasis group have any effects on the noncanonical roles of RAD51?

2.                    Figure 1. Explain the question marks. Location of the panels should be checked.

3.                    Figure 1, 2. Elaborate the sequence of events initiated upon DNA damage. Factors involved in the DNA double strand break (DSB) formation (by different intrinsic (such as by SPO11 in meiosis) and extrinsic factors (radiation, DNA damaging agents, chemical cross-linkers etc.)), primary end resection, ssDNA binding of RPA, RPA dissociation and RAD51 foci formation along with its accessory proteins should be illustrated.

4.                    A short note on the interaction of MCM8-MCM9 complex in RAD51 recruitment may be useful in context of cancer.     

5.                    Are non-HR pathways solely responsible for the repair of damaged DNA in case of RAD51 mutants or defective loaders? Does that increase the mutational burden?

6.                    Figure 1, 2. Some of the texts within the figures and sub-figures are superimposed and hard to perceive. Figure texts and titles should be clearly represented.

7.                    Figure 2, 5, 6. Explain the question marks.

8.                    Elaborate how RAD51 accessory proteins (BRCA1, BRCA2 etc.) and apical kinase proteins (ATM, ATR, CHK1, CHK2 etc.) interact and impact on the activity of RAD51 in different cancers (such as familial breast cancer and ovarian cancer etc.)?

9.                    What is the rationale for the over-expression of RAD51 in tumors?

10.                Is there any non-recombinogenic activities of RAD51 in the protection of stressed forks in terms of the RAD51 separation of function mutant (Rad51-II3A) which can form the nucleoprotein filament?

11.               Elaborate the role of RAD54 and other RAD51 accessory proteins to alleviate stalled replication forks.

12.               Does posttranscriptional R-loop forms randomly in trans when the transcript binds DNA at a distant locus?

Author Response

 Specific Concerns: 

"1. Does RAD54 (RAD54L in human) and other RAD52 epistasis group have any effects on the noncanonical roles of RAD51? "

The RAD52 epistasis group contains about ten proteins that have very different activities and therefore functions in DNA repair. RAD50, MRE11, NBS1 are part of a nuclease complex, which in addition is involved in DSB signaling, RAD52 and RAD59 have pro-annealing activities, RAD54 is a translocase, RAD55/RAD57 promotes RAD51 activity by stabilizing the filament. As a consequence, these proteins have very different impacts on the catalytic and non-catalytic activities of RAD51.

Alternative repairs:

  • RAD50/MRE11/NBS1 are implicated in resection and as such in alt-EJ (PMID: 19633668)
  • RAD52 and RAD59 catalyzes the annealing of complementary strands (SSA) (PMID: 10567339)
  • RAD54 inhibits SSA in mouse (PMID: 10757799) but has no impact in cerevisiae (PMID: 8849880)
  • RAD55/RAD57 has no impact on SSA in cerevisiae (PMID: 8849880)
  • hRAD52 inhibits the repair by Pol Theta mediated end joining, a sub-pathway of A-EJ (PMID: 34616022)

Replication forks reversal/protection:

  • MRE11 initiates replication fork degradation (PMID: 29038425)
  • As a translocase, RAD54 promotes replication fork reversal (PMID : 21097884)
  • Mammalian RAD52 prevents excessive replication fork reversal (PMID: 21097884) but, together with the PTIP complex, facilitates MRE11 targeting to stalled forks and then fork resection (PMID: 29038466).

Post replicative repair:

  • RAD54 can catalyze both regression and restoration of model replication forks through its branch migration activity, but shows strong bias toward fork restoration (PMID: 21097884)
  • RAD52 and RAD59 are required for post replicative repair (inverted-repeat recombination by a strand-annealing mechanism on replication blocks) (PMID: 21317047)
  • homologous recombination factors Rad55 and Rad57 (RAD51 paralogs in yeast), but not Rad59, are required for the formation of template switch intermediates (MMS damage in cerevisiae). PMID: 21085632

R-loop and CMM:

To our knowledge, no information is available on the putative impact of the members of the RAD52 epistasis group on R-loop formation /removal and CMM.

Note that human and yeast RAD52 show strong differences and functions. For instance, RAD51 is loaded by RAD52 in yeast but mainly by BRCA2/PALB2 in mammals.

We acknowledge that the question is of interest but it would deserve a full-length review. We think that adding and discussing these information would blur the message of the manuscript, which focusses on the non-catalytic activities of RAD51 itself (there is already a lot to describe and discuss, and this is the first review on this topic).

"2.  Figure 1. Explain the question marks. Location of the panels should be checked.

  1. Figure 1, 2. Some of the texts within the figures and sub-figures are superimposed and hard to perceive. Figure texts and titles should be clearly represented
  2. Figure 2, 5, 6. Explain the question marks."

This should result from formats problems and compatibilities between the application versions. We do not see this in none of the computers of our lab. We have changed the format of the figure and we hope that these technical problems have disappeared.

  1. Figure 1, 2. Elaborate the sequence of events initiated upon DNA damage. Factors involved in the DNA double strand break (DSB) formation (by different intrinsic (such as by SPO11 in meiosis) and extrinsic factors (radiation, DNA damaging agents, chemical cross-linkers etc.)), primary end resection, ssDNA binding of RPA, RPA dissociation and RAD51 foci formation along with its accessory proteins should be illustrated.

We have modified the figure 1, showing this on the figure.

  1. A short note on the interaction of MCM8-MCM9 complex in RAD51 recruitment may be useful in context of cancer.     

We thank the reviewer for this interesting and useful comment. In the revised version we have added the following sentence:

“Also, MCM8 and MCM9 that are implicated in prereplication complex (pre-RC) formation and DNA replication elongation and aids in normal replication fork progression, favor recruitment of BRCA1 and Rad51 upon fork stalling to protect them from excessive degradation [59].” 

  1. Are non-HR pathways solely responsible for the repair of damaged DNA in case of RAD51 mutants or defective loaders? Does that increase the mutational burden?

In absence of RAD51 or RAD51 loaders HR cannot operate; therefore, any putative repair pathway would be automatically a non-HR pathway.

We have proposed that the choice of the repair mechanism occurs in two steps: the first step is a choice between end protection (e.g. canonical-NHEJ) and end resection. The second step occurs after resection, between HR, SSA and A-EJ. In agreement with this concept, inactivation of HR in breast tumors leads to the accumulation of deletions (with an increased use of microhomologies) which are hallmarks of the increased activity of A-EJ (PMID: 27135926).

Repression of the strand exchange activity of RAD51 results in increased mutagenesis upon UV-exposure, likely through the channeling toward TLS for the bypass the UV damages PMID: 12037689. Base substitutions are also characteristic of BRCA-mutated tumors, probably as a consequence of the increased TLS when replication stalling is not manageable by HR (PMID: 36755961).

No mutational signature was determined for RAD51 mutants as they are not found in cancers and then no genomic data is available.

In the revised version we have added the following sentences:

“In agreement with this concept, inactivation of HR in breast tumors leads to the accumulation of deletions (with an increased use of microhomologies) which are considered as hallmarks of the increased activity of A-EJ [41]. Moreover, repression of the strand exchange activity of RAD51 results in increased mutagenesis upon UV-exposure, likely through the channeling toward TLS for the bypass the UV damages [42]. Base substitutions are also characteristic of BRCA-mutated tumors, probably as a consequence of the increased TLS when replication stalling is not manageable by HR [43]".

  1. Elaborate how RAD51 accessory proteins (BRCA1, BRCA2 etc.) and apical kinase proteins (ATM, ATR, CHK1, CHK2 etc.) interact and impact on the activity of RAD51 in different cancers (such as familial breast cancer and ovarian cancer etc.)?

These points are indeed very interesting but far from the scope of the present review which focuses on the non-canonical functions of RAD51 in terms of molecular mechanisms (and we already described and discussed a lot of points) on an academic point of view, and not so much on the clinical aspects. We believe that adding these descriptions and discussing them would dilute too much the message of this review, which is the first one on this topic.

  1. 9. What is the rationale for the over-expression of RAD51 in tumors?

Cancer cells are highly proliferative with an important and very active metabolism: this creates oxidative and replicative stresses that increase the reliance on HR for proliferation and survival.

In addition, tumors may have been treated with DNA damaging agents, leading to the selection of cells resistant to chemical and radiation (i.e. with highly active repair systems, notably HR).

  1. Is there any non-recombinogenic activities of RAD51 in the protection of stressed forks in terms of the RAD51 separation of function mutant (Rad51-II3A) which can form the nucleoprotein filament?

We have described in the text non-recombinogenic activities of RAD51 involved in forks protection. The RAD51 II3A mutant is deficient for strand exchange but has non-recombinogenic functions. Activities of the RAD51 II3A mutant are described in the text:

“Consistent with a dissociation of function between strand exchange (canonical role) and fork reversal activities, a mutant form of RAD51 that is unable to catalyse strand exchange, RAD51-II3A, is capable of promoting fork reversal [60]”, and “Reciprocally, RAD51-II3A, which forms filaments but is deficient in strand exchange, protects against resection [60] ”.

However, as mentioned in the manuscript, RAD51-II3A is deficient for some non-recombinogenic functions of RAD51: “RAD51-II3A, a mutant RAD51 protein with no strand invasion activity, cannot drive the recruitment of TERRA to telomeres [87] ”.

  1. Elaborate the role of RAD54 and other RAD51 accessory proteins to alleviate stalled replication forks.

As mentioned above (see reply to comment #1), RAD54 is a translocase and as such catalyzes fork reversal (PMID: 21097884).

Several RAD51 accessory proteins like BRCA1, FANCD2, and the RAD51B/RAD51C/RAD51D/XRCC2 subcomplex, contribute to protection of stressed forks by stabilizing RAD51 nucleofilaments. The CX3 subcomplex is dispensable for fork reversal, but mediates efficient restart of reversed forks (PMID: 32669601, PMID: 22789542, PMID: 26354865).

The human RAD51 mediator BRCA2 protects arrested forks against resection but mammalian RAD52, together with the PTIP complex, facilitate MRE11 targeting to stalled forks through unknown mechanisms (PMID: 29038466).

Then the relative contributions of all these factors are important but above all very different. Even if we agree that the question is of great interest, we believe that inserting this information in the manuscript would dilute and confuse the topic of the review, which is the non-canonical functions of RAD51 itself (and not the non-canonical functions of all HR related factors).

  1. Does posttranscriptional R-loop forms randomly in trans when the transcript binds DNA at a distant locus?

Trans R-loops do not bind randomly but bind to a homologous locus, which has been specified in the text in heading 4.1.

Reviewer 2 Report

Comments to the Author(s)

TitleNoncanonical Roles of RAD51

The effect of RAD 51 gene on HR has been widely studied, but it plays a less atypical role in cell survival and value added and cannot be ignored. This review further summarizes and discusses the different non-canonical roles of RAD 51, which helps us to investigate more deeply the prominent role highlighted in genomic plasticity. The overall thinking is clear, logical is reasonable, close to clinical, and has high research value. Therefore, I suggest that this manuscript be received after some modifications.

Comments needed to be focused on:

1.    In the introduction section, the author has less foreshadowing for the atypical role of RAD51 in DSB repair and DNA replication in the following text, and suggests adding some introduction to the following content.

2.      Although this article has written a lot about the different atypical effects of RAD51 in clinical aspects, it has not mentioned the direction of the next research plan of RAD51 in the article, so the readers cannot see the profound research significance of this review.

3.    In line 16, you propose the theory of "RAD51 paradox", and then propose that RAD51 can perform other atypical roles based on this part. I think this inference is not supported by sufficient statements.

4.     At the end of each heading, it is recommended that you provide a small summary of the content of each part of the heading, summarizing the central idea of the content of the paragraph. This will not only enhance the reading efficiency of the reviewer, but also reflect the organization and summary of this article.

5.      .The fourth part is not closely related to clinical practice. It is suggested that the author add clinical cases about RAD51 promoting the formation and elimination of R-loops.

6.      Germline pathogenic variation has been found in CMM syndrome. Can the author try to make some assumptions about the relationship between RAD51 mutation and tumorigenesis?

7.    Figure 1,2,6 all appear in the frame of the question mark, is the mapping error? Please explain the reason why it appears.

8.    The blue-colored DNA, purple-colored RAD51 protein, and red arrows mentioned in the Figure 5 explanation are not shown in the diagram.

9.    The second part focuses on the atypical role of RAD511 in protecting A-EJ and SSA through the chain exchange activity of non-catalytic ssDNA, but there are too many beddings for DSB, it is recommended to refine it.

10.  Line-397, whether more ' - '.

Author Response

Comments needed to be focused on:

  1. In the introduction section, the author has less foreshadowing for the atypical role of RAD51 in DSB repair and DNA replication in the following text, and suggests adding some introduction to the following content.

We have added a paragraph at the end of the introduction addressing these points.

  1. Although this article has written a lot about the different atypical effects of RAD51 in clinical aspects, it has not mentioned the direction of the next research plan of RAD51 in the article, so the readers cannot see the profound research significance of this review.

We hope that the text led the readers think that investigating noncanonical functions of RAD51 is as important as investigating recombinogenic functions in terms of maintenance of genome stability, tumor development. This is particularly important in the course of RAD51 inhibitors development.

To support this notion, we have added the following sentence in the concluding remarks of the revised version:

“RAD51 is a pharmacological target, and active research is performed to develop RAD51 inhibitors [117]. Consequently, taking into account the impact of these inhibitors on RAD51-noncanonical functions should be highly valuable: molecules that can affect all RAD51 functions, or in contrast that are able to selectively inhibit one or the other of the different functions of RAD51 should allow enlarging the panel of therapeutic strategies, and to adapt them to different clinical cases.”

  1. In line 16, you propose the theory of "RAD51 paradox", and then propose that RAD51 can perform other atypical roles based on this part. I think this inference is not supported by sufficient statements.

We already have proposed and extensively discussed the theory of the "RAD51 paradox", which largely is supported by numerous data and observations (PMID: 34316706). The theory of the "RAD51 paradox" is not the prime topic of our present manuscript, which focus on the noncanonical functions of RAD51 (although functions might be connected to the "RAD51 paradox"). To summarize the "RAD51 paradox" is based upon the facts that: 1) RAD51 plays the pivotal roles in HR; 2) the other proteins are mediator/accessory proteins; 3) despite the prime role RAD51 causal mutations have not been identified in cancer, including in breast and ovary cancer, in contrast with its partners (mediators and accessory proteins): 5) therefore, RAD51 is not included in the list of genetic screens in clinic (in contrast with the mediators and accessory genes). This constitutes a strong intriguing paradox. This is confirmed in Fanconi anemia in which patients bearing RAD51 mutations did not (for the moment) develop cancer (in contrast with patients bearing mutations in BRCA1 or BRCA2). In the congenital mirror movement syndrome many patients bear RAD51 mutations, but this syndrome is not associated with cancer predisposition. Therefore, there is a substantial amount of data supporting the concept of "RAD51 paradox"  In the revised version, we have developed in the introduction, the hypotheses that could connect the "RAD51 paradox" with the noncanonical roles of RAD51, as suggested by the reviewer.

  1. At the end of each heading, it is recommended that you provide a small summary of the content of each part of the heading, summarizing the central idea of the content of the paragraph. This will not only enhance the reading efficiency of the reviewer, but also reflect the organization and summary of this article.

In the revised version we have added some sentences to summarize each paragraph.

  1. The fourth part is not closely related to clinical practice. It is suggested that the author add clinical cases about RAD51 promoting the formation and elimination of R-loops.

This review is not a clinical but an academic review.

  1. Germline pathogenic variation has been found in CMM syndrome. Can the author try to make some assumptions about the relationship between RAD51 mutation and tumorigenesis?

The CMM syndrome is not associated with cancer predisposition, as stated in the text. More generally, RAD51 mutations are not associated with tumorigenesis (see above and PMID: 34316706)

  1. Figure 1,2,6 all appear in the frame of the question mark, is the mapping error? Please explain the reason why it appears.

This should result from format problems and compatibilities between the application versions. We do not see this in none of the computers of our lab. We have changed the format of the figure and we hope that these technical problems have disappeared.

  1. The blue-colored DNA, purple-colored RAD51 protein, and red arrows mentioned in the Figure 5 explanation are not shown in the diagram.

This has been corrected in the revised version.

  1. The second part focuses on the atypical role of RAD511 in protecting A-EJ and SSA through the chain exchange activity of non-catalytic ssDNA, but there are too many beddings for DSB, it is recommended to refine it.

Sorry but we do not understand this comment.

  1. Line-397, whether more ' - '.

This has been corrected in the revised version.

Reviewer 3 Report

With this review, the authors aim to collect data regarding noncanonical roles of RAD51. I found this paper well orgnized and quite complete. In my opinion, some minor revisions should be made in order to improve this work.

In particular:

- references shoul be checked very carefully, expecially paragraph 4.1. "RAD51 promotes R-loop formation" where a few papers are not correctly cited;

- authors should also correct some details in the figures, since overlapping writings and different symbols are present on figures 1, 2 and 6.

- legends have to be checked. For example, see "Left" (at line 75- figure1) instead of right and "red arrows"  (at line 320- figure 5) which are not drawn in the figure. Moreover, Figure 4 should be better explained.

Author Response

- references should be checked very carefully, especially paragraph 4.1. "RAD51 promotes R-loop formation" where a few papers are not correctly cited;

We are sorry for this confusion. This has been corrected

- authors should also correct some details in the figures, since overlapping writings and different symbols are present on figures 1, 2 and 6.

This should result from format problems and compatibilities between the application versions. We do not see this in none of the computers of our lab. We have changed the format of the figure and we hope that these technical problems have disappeared.

- legends have to be checked. For example, see "Left" (at line 75- figure1) instead of right and "red arrows"  (at line 320- figure 5) which are not drawn in the figure. Moreover, Figure 4 should be better explained.

This has been corrected in the revised version.

Reviewer 4 Report

The review by Thomas et. al. on the noncanonical roles of RAD51 makes for an informative read. The authors have provided an in-depth account of the current understanding of RAD51 and its roles beyond HR. However, I believe the review would benefit from addressing the following comments:

·      In the non-canonical roles of RAD51, it is imperative to mention HR via the alternate mediator of RAD51 loading, RAD52, especially in the BRCA-deficient setting. This could be woven into the introduction or the model/body of the text as the authors see fit.

·      In line 87-88, cite the original references that indicate that “RAD51 is overexpressed in tumours”

·      Discuss how the paper by Lafuente-Barquero et al (https://doi.org/10.7554/eLife.56674) fits in with the role of RAD51 at R-loops.

·      The review would benefit from a summary figure listing all the non-canonical functions of RAD51 other than HR.

Author Response

  • In the non-canonical roles of RAD51, it is imperative to mention HR via the alternate mediator of RAD51 loading, RAD52, especially in the BRCA-deficient setting. This could be woven into the introduction or the model/body of the text as the authors see fit.

This is mentioned in the legend of figure 1 and the following sentence has been added in the text of the revised version :

“However, BRCA2 deficient cells can promote homology-directed DSB repair (HDR) in a RAD52-dependant manner and synthetic lethality between BRCA2 and RAD52 has been reported [13,14].”

  • In line 87-88, cite the original references that indicate that “RAD51 is overexpressed in tumours”

We initially chose to cite a review we wrote to avoid too many references but added the following references in the modified version of the manuscript.

  • Tennstedt,P., Fresow,R., Simon,R., Marx,A., Terracciano,L., Petersen,C., Sauter,G., Dikomey,E. and Borgmann,K. (2013) RAD51 overexpression is a negative prognostic marker for colorectal adenocarcinoma. J. Cancer, 132, 2118–2126.
  • Maacke,H., Jost,K., Opitz,S., Miska,S., Yuan,Y., Hasselbach,L., Luttges,J., Kalthoff,H. and Sturzbecher,H.W. (2000) DNA repair and recombination factor Rad51 is over-expressed in human pancreatic adenocarcinoma. Oncogene, 19, 2791–2795.
  • Maacke,H., Opitz,S., Jost,K., Hamdorf,W., Henning,W., Kru ̈ ger,S., Feller,A.C., Lopens,A., Diedrich,K., Schwinger,E. et al. (2000) Over-expression of wild-type Rad51 correlates with histological grading of invasive ductal breast cancer. J. Cancer, 15, 907–913.
  • Qiao,G.B., Wu,Y.L., Yang,X.N., Zhong,W.Z., Xie,D., Guan,X.Y., Fischer,D., Kolberg,H.C., Kruger,S. and Stuerzbecher,H.W. (2005) High-level expression of Rad51 is an independent prognostic marker of survival in non-small-cell lung cancer patients. J. Cancer, 93, 137–143.
  • Welsh,J.W., Ellsworth,R.K., Kumar,R., Fjerstad,K., Martinez,J., Nagel,R.B., Eschbacher,J. and Stea,B. (2009) Rad51 protein expression and survival in patients with glioblastoma multiforme. J. Radiat. Oncol. Biol. Phys., 74, 1251–1255.
  • Li,Y., Yu,H., Luo,R.Z., Zhang,Y., Zhang,M.F., Wang,X. and Jia,W.H. (2011) Elevated expression of Rad51 is correlated with decreased survival in resectable esophageal squamous cell carcinoma. Surg. Oncol., 104, 617–622.

  • Discuss how the paper by Lafuente-Barquero et al (https://doi.org/10.7554/eLife.56674) fits in with the role of RAD51 at R-loops.

We have discussed this paper in chapter 4.1 of the revised version.

  • The review would benefit from a summary figure listing all the non-canonical functions of RAD51 other than HR.

We thank the reviewer for this good idea. In the revised version have added a graphical abstract summarizing these activities.

Round 2

Reviewer 2 Report

Comments to the Author(s)

TitleNoncanonical Roles of RAD51

In the first revision, you have responded in detail to most of the comments I have made. As for the ninth opinion, you do not understand the place and did not modify it. What I'm trying to say about this piece of advice is the second part of the review focuses on the atypical role of "RAD511 to protect A-EJ and SSA by strand exchange activity of non-catalytic ssDNA", but there are too many descriptions for the DSB ahead, so it is recommended to refine this part. Please revise the ninth suggestion further according to my suggestion.

Author Response

1- The reviewer wrote:
"RAD511 to protect A-EJ and SSA by strand exchange activity of non-catalytic ssDNA"

RAD51 (and not RAD511 but it is probably a typo) prevents A-EJ and SSA, which is the complete opposite of what the reviewer wrote.

Then, with regards to the end of the sentence: strand exchange activity IS a catalytic activity sensu stricto; as such it cannot be promoted by a non-catalytic molecule. Incidentally "non-catalytic ssDNA » doesn't make more sense as what bears the catalytic or non-catalytic activity is the RAD51 filament and not the ssDNA, which is obviously devoid of catalytic activity.

2- Then the commentary about DSB is odd: " but there are too many descriptions for the DSB ahead,"

The heading of this section is:" Noncatalytic roles of RAD51 in DSB repair". Therefore, DSB is the central point and its clear description is necessary. It is important to point out that DSB is a prominent type of DNA damage: it is the most toxic DNA damage and DSBs are necessary in important fundamental processes such as meiosis and the establishment of the immune system. Therefore, the study of DSB repair is an essential field in genetics.

We have presented the canonical role of RAD51. Indeed, as this review presents and discusses the "noncanonical roles of RAD51 » then it is imperative to define what is canonical, before discussing what is not. 

Finally, rev#1 ask us to detail more DSBs and to add the sources of DSBs (Point 3 of rev#1: " Elaborate the sequence of events initiated upon DNA damage. Factors involved in the DNA double strand break (DSB) formation …". Then we feel that it is difficult to fulfil all reviewers’ requests as they are contradictory and we believe comments by rev#1 were wise.

However, in an attempt to address what we understand of the comment we have modified the organization of the manuscript: we have removed the description of the canonical role of RAD51 from the introduction section and we have created a novel section (section #2) describing these functions. We hope that these changes make the review clearer.

Moreover, we have found a recent publication that deserves to be quoted in our review; therefore we have added the following sentences in the revised version: "Recently, it has been shown that RAD51 is implicated in the protection of T cells against senescence, promoting long term immunological memory [46]. Indeed, antigen presenting cells (APC) were found to transfer telomeres sequences to T cells, maintaining telomere length of the recipient T cells and thus protecting them from senescence. This telomere transfer mechanism acts through an intercellular process that requires synapsis between APCs and T cells and the production of extracellular vesicles by APCs. These vesicles contain telomeres fragments, and HR proteins BRCA1, BRCA2 and RAD51. The production of telomeres-containing vesicles is independent of RAD51 but the content in telomeric ssDNA in vesicles, and the lengthening of telomeres in recipient T cells do require RAD51 [46]. Deciphering the respective parts of the catalytic strand invasion activity versus the noncanonical ssDNA-protection function by RAD51 in this process remains to be investigated".

We hope this novel revised version will be suitable for publication.